# Metabolic Profile and Metabolite Analyses in Extreme Weight Responders to Gastric Bypass Surgery

**DOI:** 10.3390/metabo12050417

**Published:** 2022-05-06

**Authors:** Charlotte M. Fries, Sven-Bastiaan Haange, Ulrike Rolle-Kampczyk, Andreas Till, Mathis Lammert, Linda Grasser, Evelyn Medawar, Arne Dietrich, Annette Horstmann, Martin von Bergen, Wiebke K. Fenske

**Affiliations:** 1Department of Endocrinology, Diabetes and Metabolism, University Hospital Bonn, Venusberg-Campus 1, 53127 Bonn, Germany; a.till@uni-bonn.de (A.T.); wiebke.fenske@ukbonn.de (W.K.F.); 2Department of Molecular Systems Biology, Helmholtz Centre for Environmental Research GmbH-UFZ, Permoserstraße 15, 04318 Leipzig, Germany; sven.haange@ufz.de (S.-B.H.); ulrike.rolle-kampczyk@ufz.de (U.R.-K.); martin.vonbergen@ufz.de (M.v.B.); 3Department of Neurology, Max Planck Institute for Human Cognitive and Brain Sciences, Stephanstraße 1a, 04103 Leipzig, Germany; mathislammert@googlemail.com (M.L.); linda.grasser@posteo.de (L.G.); medawar@cbs.mpg.de (E.M.); horstmann@cbs.mpg.de (A.H.); 4Department of Visceral and Metabolic Surgery, University Hospital Leipzig, Liebigstraße 18, 04103 Leipzig, Germany; arne.dietrich@medizin.uni-leipzig.de; 5Department of Psychology and Logopedics, Faculty of Medicine, University of Helsinki, Haartmaninkatu 3, 00290 Helsinki, Finland; 6German Centre for Integrative Biodiversity Research (iDiv) Halle-Jena-Leipzig, Puschstraße 4, 04103 Leipzig, Germany; 7Faculty of Life Sciences, Institute of Biochemistry, University of Leipzig, Brüderstraße 34, 04103 Leipzig, Germany

**Keywords:** bile acids, bariatric surgery, Roux-en-Y gastric bypass, weight response, gut microbiota

## Abstract

Background: Roux-en-Y gastric bypass (RYGB) surgery belongs to the most frequently performed surgical therapeutic strategies against adiposity and its comorbidities. However, outcome is limited in a substantial cohort of patients with inadequate primary weight loss or considerable weight regain. In this study, gut microbiota composition and systemically released metabolites were analyzed in a cohort of extreme weight responders after RYGB. Methods: Patients (*n* = 23) were categorized based on excess weight loss (EWL) at a minimum of two years after RYGB in a good responder (EWL 93 ± 4.3%) or a bad responder group (EWL 19.5 ± 13.3%) for evaluation of differences in metabolic outcome, eating behavior and gut microbiota taxonomy and metabolic activity. Results: Mean BMI was 47.2 ± 6.4 kg/m^2^ in the bad vs. 26.6 ± 1.2 kg/m^2^ in the good responder group (*p* = 0.0001). We found no difference in hunger and satiety sensation, in fasting or postprandial gut hormone release, or in gut microbiota composition between both groups. Differences in weight loss did not reflect in metabolic outcome after RYGB. While fecal and circulating metabolite analyses showed higher levels of propionate (*p* = 0.0001) in good and valerate (*p* = 0.04) in bad responders, respectively, conjugated primary and secondary bile acids were higher in good responders in the fasted (*p* = 0.03) and postprandial state (GCA, *p* = 0.02; GCDCA, *p* = 0.02; TCA, *p* = 0.01; TCDCA, *p* = 0.02; GDCA, *p* = 0.05; GUDCA, *p* = 0.04; TLCA, *p* = 0.04). Conclusions: Heterogenous weight loss response to RYGB surgery separates from patients’ metabolic outcome, and is linked to unique serum metabolite signatures post intervention. These findings suggest that the level of adiposity reduction alone is insufficient to assess the metabolic success of RYGB surgery, and that longitudinal metabolite profiling may eventually help us to identify markers that could predict individual adiposity response to surgery and guide patient selection and counseling.

## 1. Introduction

Management of obesity and its extensive disease burden is one of the greatest challenges of modern medicine. Bariatric surgery remains the most effective therapeutic approach for morbid obesity, as well as for several associated co-morbidities, including type 2 diabetes mellitus (T2D), dyslipidemia, hypertension, non-alcoholic fatty liver disease (NAFLD) and cardiovascular diseases [1,2,3,4,5,6,7]. Roux-en-Y gastric bypass (RYGB) and vertical sleeve gastrectomy (VSG) are the two most frequently performed surgical interventions worldwide for sustained weight loss and improved glucose metabolism [8]. While both procedures stand out amongst currently available weight loss strategies by their short- and long-term effectiveness to reduce adiposity and improve obesity-related systemic metabolic disorders, the individual patient response to surgery varies widely and a sizeable proportion of patients struggles with a relatively poor achievement of primary weight loss or even a pronounced weight regain after initial adiposity reduction [3,8,9,10,11,12]. As the magnitude of weight loss is closely related to the improvement of obesity-related comorbidities and patient satisfaction with the intervention [13,14,15,16], insufficient weight loss is the most common reason for revisional bariatric surgery today [17].

Although a universally acknowledged definition for “insufficiency” of weight loss or “weight regain” is lacking [18,19], it is estimated that prevalence varies between 15–30% after RYGB [10,20,21,22,23] and may be the result of a complex interaction between multiple genetic traits [24,25], as well as psychological, behavioral, nutritional, environmental and surgery-related factors [26,27].

It is of note that the mechanisms by which the weight loss surgical procedures achieve sustained weight reduction and metabolic improvements beyond adiposity reduction remain incompletely understood [28,29,30]. Latest insights from others [31] and our own studies [32,33,34] indicate that the altered intestinal environment after RYGB-like gut reconfiguration induces remarkable top-down effects on the composition, diversity and metabolic activity of the gut microbiota, which may play an important role in the beneficial effects of the surgery. While several lines of evidence point to a decisive role for the altered gut microbiome [35] and increased bile acid abundance [36,37,38,39] in supporting the effects of bariatric surgery on sustained adiposity reduction and metabolic improvements, the parallel dramatic changes in body weight and eating behavior essentially limit this interpretation. Moreover, the specific role of the gut microbiome and bacteria-derived metabolites in variable weight loss response to bariatric surgical intervention, as well as its consequences on sustained metabolic control, remain poorly understood.

The current study was designed to profile the composition of the gut microbiota and the circulating metabolomic signature in patients with variable weight loss response to RYGB surgery and to determine associations with weight loss-independent beneficial metabolic outcomes after surgery.

## 2. Results

### 2.1. Study Cohort and Metabolic Profile

Patients after RYGB surgery were identified via the local Adiposity Research database by screening for the 5% best and worst weight loss responders (defined by excess weight loss (EWL)) with comprehensive metabolic characterization and available follow-up data for at least 24 months after bariatric surgery. Inclusion criteria were a body mass index (BMI) between 40–60 kg/m^2^ before RYGB and a minimum time span of two years since bariatric surgery. Exclusion criteria were acute neurological or psychiatric disorders, alcohol or drug abuse, prior neurosurgical procedures or head trauma. After study enrolment, patients were invited for follow up visit where they received a standardized test meal after an overnight fasting period, answered questionnaires addressing amongst others eating behavior and diet preferences (see Section 4.1) and underwent anthropometric measurements. Blood samples were collected before and at several time points after the test meal, and patients donated feces for microbiota and metabolome analysis.

Table 1 shows the anthropometric data and metabolic profile of the study cohort (*n* = 23 subjects) before surgery and at follow up. Patients were predominantly female in both groups (64% in good and 75% in bad responders). Mean age (± SD) at test date (52.9 ± 9.5 years in good and 54.1 ± 10.6 years in bad responders) and time after surgery (4.3 ± 1.2 in good and 4.6 ± 1.5 years in bad responders) did not differ between groups.

Mean body mass index (BMI ± SD) before surgery was higher in the bad responder group (52.7 ± 6.6 vs. 46.5 ± 7.5 kg/m^2^; *p* = 0.04). BMI at follow up was 47.2 ± 6.4 kg/m^2^ in the bad vs. 26.6 ± 1.2 kg/m^2^ in the good responder group with a corresponding change in BMI points of −5.5 ± 3.9 vs. −19.8 ± 6.7 and excess weight loss of 19.5 ± 13.3% and 93 ± 4.3%, respectively.

As we previously described [34], bad responders showed higher eating restraint scores as well as eating, weight and shape concerns, whereas self-reported sugar and fat intake was similar (Table 1). The vast majority of patients in both groups reported reduced food intake after bariatric surgery, which were 10 patients (91%) in the good and 10 patients (83%) in the bad responder group. Two subjects (17%) in the bad responder group, but only one subject (9%) in the good responder group reported binge eating. Questionnaires evaluating emotional eating and chronic stress exposed no inter-group differences.

Metabolic measures included serum liver enzyme levels, lipid levels and glucose homeostatic parameters, which showed no differences at baseline. At follow up, good responders had lower median (IQR) fasting glucose levels (5 (4.6–5.3) vs. 5.4 (5.2–8.5) mmol/L; *p* = 0.02), hemoglobin A1c concentration (5 (4.8–5.6) vs. 5.7 (5.2–6.6)%, *p* = 0.03), γ-glutamyl transferase (γGT) (0.21 (0.13–0.42) vs. 0.41 (0.25–1.20) μkat/L; *p* = 0.05) and triglyceride levels (0.94 (0.63–1.2) vs. 1.3 (1.2–1.3) mmol/L; *p* = 0.02).

However, both groups showed marked improvement of metabolic profiles and group differences were lost for all parameters except for high density lipoprotein (HDL) (0.6 ± 0.27 in good vs. 0.28 ±0.3 mmol/L in bad responders; *p* = 0.02) when absolute changes from baseline were compared. Pre-surgery elevated C-reactive protein (CRP) levels were markedly lowered in both responder groups at follow up, while levels were lower in good responders (0.3 (0.3–0.3) vs. 2.0 (1.3–13) mg/L, *p* < 0.01) and suppressed below detection limit in all but one patient.

Figure 1 shows the proportion of patients with T2D and hemoglobin A1c categories of both groups before and after RYGB. Four patients (36%) in the good responder group were diabetic before intervention and all experienced remission (*n* = 2) or improvement (*n* = 2) of their glycemic control at follow up [40]. In the bad responder group, six patients (50%) were diabetic, of which two patients were in remission at follow up, with one patient experiencing improvement, and three patients experienced no change or worsening of their glycemic control.

### 2.2. Hunger and Satiety Rating and Gut Hormone Release during Standardized Mixed-Meal Test

Extreme responders to RYGB surgery received at follow up a mixed-meal test by ingesting a 125 mL liquid meal containing 300 kilocalories (see Section 4.1 for detailed nutrient specifications). Figure 2 shows ratings for hunger, satiety and palatability of the meal (Figure 2A–E) as quantified by visual analogue scale (VAS), as well as corresponding concentrations of gut hormone release (Figure 2D–J) during the mixed-meal test. Notably, both responder groups ranked hunger, satiety and palatability levels similarly during the standard test meal. Furthermore, gut hormone levels did not differ significantly between both groups in the fasted state. Only leptin levels were higher in the bad responder group (35,739 (24,287–59,467) vs. 3167 (1321–5242) pg/mL; *p* < 0.0001).

Moreover, although good responders showed a trend towards higher stimulated glucagon-like peptide−1 (GLP-1) and peptide YY (PYY) release and a reduced insulin release (Figure 2K–M) after the test meal compared to bad weight loss responders, differences were not statistically significant. Plasma ghrelin and leptin levels were further baseline corrected to account for weight differences and showed a statistically not-significant trend towards a more pronounced ghrelin suppression in good responders, and no discernable secretion profile for leptin after the test meal in both responder groups (Figure 2N,O).

### 2.3. Fecal Microbiota Composition and Metabolomics

#### 2.3.1. Fecal Microbiota Composition and Bacterial Metabolites

To investigate possible differences in the long-term effects of RYGB surgery on the gut microbiota in extreme weight loss responders, fecal samples of both weight loss responder groups were profiled for microbiota composition and bacteria-linked metabolites as pivotal mediators of host–microbiota communication. Microbiome taxonomic structure of good responders and bad responders showed no differences in richness or alpha-diversity based on Shannon index calculation (Appendix A). In addition, beta-diversity, distribution of microbial families in each sample and abundance of single genera did not differ (Appendix A) between both responder groups. Additionally, despite obviously clear differences in adiposity development between the two groups, targeted metabolomics in feces yielded no differences in amino acids, biogenic amines, acyl carnitines and hexose (Appendix A). Out of ten short chain fatty acids (SCFA), valerate was the only metabolite found to be more abundant in bad responders compared to good weight loss responders (Appendix A; *p* = 0.04).

#### 2.3.2. Circulating Metabolites and Bile Acids

Circulating metabolites and bile acid species were quantified at follow up both in the fasted and postprandial state at 0, 30, 60 and 120 min during the mixed-meal test.

Fasting serum SCFA concentration and postprandial release in response to the standardized test meal were quantifiable for propionate, valerate and acetate. Propionate showed higher abundance at fasted and stimulated state in good responders (*p* = 0.0001), whereas postprandial valerate response was higher (*p* = 0.04) in bad responders (Figure 3A,B). Acetate levels did not differ at either time point. Additionally, serum amino acids (AA) and biogenic amines showed no differences between groups neither at fasted state nor under test meal-stimulated conditions (Figure 3C–E). SCFA and AA levels were present in comparable proportions, indicating that intestinal fermentation, and thereby energy harvest, did not differ between both responder groups.

Interestingly, we found different modulation of bile acid concentration in extreme responders to RYGB surgery both at the fasted and stimulated states. Fasting serum bile acid concentration showed higher abundance of both conjugated primary and secondary bile acids in good responders (Figure 4A,B; *p* = 0.03 for cumulative conjugated bile acids and primary and secondary conjugated bile acids independently in good vs. bad responders). Individual analyses of bile acid species showed significantly higher concentrations, particularly of glycochenodeoxycholic acid (GCDCA; Figure 4D; *p* = 0.03) and glycochenodeoxycholate acid (GCDA; Figure 4D; *p* = 0.02), in the good responder group.

After ingestion of the test meal, good responders showed a clearly higher serum concentration for the conjugated primary bile acids (glycocholic acid, GCA; glycochenodeoxycholic acid, GCDCA; taurocholic acid, TCA; and taurochenodeoxycholic acid, TCDCA) and the conjugated secondary bile acids (glycochenodeoxycholate acid, GCDA; glycoursodeoxycholic acid, GUDCA; and taurolithocholic acid, TLCA). Figure 5 depicts concentrations for the statistically significant bile acids at time points 0, 30, 60 and 120 min during the mixed-meal test. Group differences were most evident at time point 30 min for most bile acids and for the four conjugated primary bile acids GCA (Figure 5A; *p* = 0.02), GCDCA (Figure 5B; *p* = 0.02), TCA (Figure 5E; *p* = 0.01) and TCDCA (Figure 5F; *p* = 0.02).

## 3. Discussion

In the management of obesity and related diseases, physicians face significant heterogeneity in response to weight loss surgery. Indeed, one of the most challenging clinical issues in personalized obesity medicine is to predict how an individual will respond to weight loss surgical intervention, and to what extent the level of weight loss contributes to metabolic reprogramming and improvement of obesity-related metabolic and cardiovascular co-morbidities.

The present study aimed to characterize a cohort of extreme weight responders to RYGB surgery as to differences in metabolic outcome, hunger and satiety sensation, as well as to metabolite profiles in the energetically stabilized postoperative state. Interestingly, patients in both groups did not significantly differ as to age, education or glycemic control before intervention. However, bad responders had higher preoperative BMI (52.7 ± 6.6 vs. 46.5 ± 7.5 kg/m^2^; *p* = 0.04) which was previously found to be predictive for poorer post-surgery weight loss [41].

Unfavorable dietary habits, and poor diet quality, have amongst other factors been associated with poor weight loss response to bariatric surgery [42,43,44]. In our cohort, self-reported food and dietary fat and sugar intake did not differ between groups nor did emotional eating or chronic stress sensation. Two subjects of the bad responder group reported binge eating compared to one in the good responder group. As previously reported [34], bad responders in our analysis did show higher scores in all four subscales (restraint, eating concern, shape concern and weight concern) of the EDE-questionnaire. While this has been associated with poorer weight loss after bariatric surgery before [38], the higher scoring could also be attributed to higher BMI itself in the poor responder group [45].

Here, our data reveal several important novel implications. First, our results indicate that postoperative adaptions of energy homeostasis largely dissociate from metabolic control after RGYB. While weight loss and caloric restriction may probably play a critical role in improved glycemic control after RYGB [28,46], our findings indicate that weight loss-independent mechanisms also appear to be involved in systemic metabolic improvements. Patients of both groups showed marked and comparable improvement in glycemic control, dyslipidemia and liver enzymes despite divergent weight loss responses. This observation is in line with previous studies which reported favorable metabolic outcomes in patients after RYGB despite “failed” weight loss [47], or rather cardiometabolic improvements across the entire spectrum on post-bariatric weight loss [48]. The controversy remains whether these improvements are indeed independent of the loss in excess weight, or the result of a minor or moderate adiposity reduction, proving sufficient to beneficially modulate metabolic outcomes [49,50]. A very elegant study has recently shown, that in a controlled weight loss intervention in obese patients 5% weight loss was sufficient to improve organ tissue insulin sensitivity and ß-cell function, while further weight loss of up to 10 to 15% is required to cause dose-dependent alterations in key adipose tissue biological pathways [50]. In view of these data, even a poor weight loss response to surgery compared to the surgical benchmark may mediate valuable and sustained metabolic benefits.

Secondly, despite clearly divergent weight loss responses to surgery, notably subjects reported no differences in hunger and satiety sensation. This is remarkable since a major part of the weight loss success of RYGB is generally attributed to sustainably lower caloric intake [33,51,52], resulting from multifaceted alterations in nutrient sensation [51]. It has been shown in rodent models that for progressive weight loss development, sensory information from the gut must reach hindbrain nucleus tractus solitarius (NTS) neurons through the celiac branch of the vagus nerve [52]. Surprisingly, it remains largely unclear which gut-derived signals precisely promote the structural and functional brain modifications after RYGB surgery. Although augmented GLP-1 release from enteroendocrine L-cells acting on peripheral and/or central GLP-1 receptors was initially considered as a major contributor of caloric restriction and post-RYGB weight loss, it has been difficult to prove the role of GLP-1 experimentally [53,54]. In line with this uncertainty and no discernible difference in hunger and appetite sensation during the test meal, we notably found a very similar fasting, as well as postprandial GLP-1 and PYY release, between both patient groups despite opposed weight loss responses. Even though a trend towards a more pronounced postprandial PYY and GLP-1 release together with a more pronounced ghrelin suppression was found in the good responder group, differences were subtle and might rather be attributed to absolute differences in glycemic control at test date [23]. While significance might have been lost due to the small sample size, our results argue against a dominant functional role of incretin hormones in sustained weight reduction after RYGB surgery. While our data argue against an elemental role of GLP-1 and PYY in long-term adiposity reduction, they do not exclude a possible and even likely functional asset for improvement in post-RYGB glycemic control [55,56,57].

Thirdly, although no differences were observed in microbial composition between good and bad weight loss responders as previously reported by others [58], metabolomics analyses revealed circumscribed intergroup differences in metabolite profiles. To our knowledge, this is the first study analyzing circulating metabolite release in relation to weight loss response to RYGB surgery. In the analyzed fecal and serum SCFAs, serum propionate was more abundant in good responders, and fecal and serum valerate levels conversely more so in bad responders. An increase of colonic propionate stimulates GLP-1 and PYY release in mice and rats [57], and reduces weight [59] and obesity-related fatty liver disease [60] in diet-induced obese mice. In overweight human individuals, propionate supplementation also showed anti-obesity effects by increasing postprandial GLP-1 and PYY release and reducing energy intake [61]. Our data support that SCFA and especially altered propionate concentrations might also play a role in more pronounced weight loss after RYGB.

In addition, we found significant differences in circulating bile acids between both groups, with increased levels of primary and secondary conjugated bile acid species in the good responder group. Interestingly, particularly the postprandial release of bile acid species was clearly increased in good responders, which was most significant for the primary conjugated bile acids species.

These cholesterol-derived molecules produced by the liver are meanwhile known for their hormone-like effects on energy and glucose metabolism [62,63] through activation of the bile acid nuclear receptor farnesoid X receptor (FXR) and the membrane-bound Takeda G protein-coupled receptor 5 (TGR5) [64,65]. A growing body of evidence has shown that bariatric surgery induces a pronounced shift in bile acid metabolism and subsequent receptor signaling, which appears to be important and necessary for the weight loss effects and glycemic control of VSG [36,66] and bile diversion to the ileum [38,67] in mice. However, human data, and especially those from RYGB intervention, are rather limited, even though increased circulating levels of bile acid species have been observed after RYGB as well [68,69], and have been linked to increased GLP-1 and reduced glucose and triglyceride levels [70,71,72]. Our data complement this picture by identifying important differences in bile acid metabolism and postprandial release of the signaling molecules. As these differences were not attributable to self-reported appetite and hunger sensation nor to strong differences in incretin hormone responses, other mechanisms of anti-obesity actions beyond GLP-1 mediated appetite suppression have to be suspected. In this context, direct effects of altered gut-derived bile acid signaling to the central nervous system might be highly relevant, and even more so as bile acids are found in the brain where their levels correlate with circulating ones [73]. Interestingly, Castellanos-Jankiewicz et al. recently demonstrated in obese mice that activation of hypothalamic TGR5 achieved a negative energy balance via modulation of food intake and energy expenditure through stimulation of the sympathetic nervous system [74]. Although circulating bile acid levels correlate with energy expenditure in healthy humans [75] and with changes in energy and substrate metabolism in obese subjects undergoing RYGB [72,76], a correlation between changes in bile acid levels with weight loss response to RYGB has never been reported and never been linked to outcome-specific differences in energy and substrate metabolism and metabolic rate. Together with the difference we found in circulating propionate levels in extreme responders, which has been shown to affect resting energy expenditure and lipid oxidation in healthy volunteers [77], our data may indicate a critical role of gut-derived metabolites in metabolic adaption and heterogeneous weight loss outcome after RYGB.

There are several limitations of our work. Firstly, the sample size is very small, which limits application of results to a broader set of patients. Secondly, the missing longitudinal design and baseline characterization of microbiota and circulating metabolites are limiting factors of this study, especially as baseline microbiota composition might be a factor influencing the outcome of bariatric surgery itself [78,79]. In addition, inter- and intraindividual variances in gut microbiota composition essentially limit interpretation of cross-sectional analyses vs. analyses that include longitudinal repeat sampling [80,81]. Lastly, the assessment of eating behavior, diet and exercise was done by self-reporting measures which might favor a social desirability bias.

Overall, our findings indicate a potential role of circulating bile acids on long term energy control. Therefore, it will be interesting for future studies to delineate the role of basal metabolic rate and individual metabolic adaption to post-surgery weight loss maintenance as a possibly powerful mediator or even an early predictor of sustained weight loss success [82,83].

## 4. Materials and Methods

### 4.1. Study Cohort and Test Date

This study was conducted at the Integrated Research and Treatment Centre for Adiposity Diseases (IFB), Department of Medicine of the University of Leipzig, Germany, and the Max Planck Institute for Human Cognitive and Brain Sciences (MPI CBS), Leipzig, Germany. The Ethical Committee of the University of Leipzig (027/17-ek) approved this study. Written informed consent was acquired prior to study participation. Subjects were identified and contacted via the IFB Adiposity Research database after screening for the 5% best and worst weight loss responders categorized by EWL at a minimum time span of 2 years after bariatric surgery. Definitions of good and bad response were based on criterion EWL that was calculated as 100 − {[(BMI_after-25_)/(BMI_before-25_)] × 100} with ideal body weight set at BMI 25 kg/m^2^. Mixed-meal tests were performed after a fasting period of 12 h using 125 mL bottles of *Nutricia Fortimel Compact*, Nutricia Milupa GmbH, Hamburg, Germany. Nutrient content is 300 kilocalories containing 12 g of protein, 12 g of fat and 37 g of carbohydrates. Blood samples were collected before and at 15, 30, 60 and 120 min after ingestion of the liquid meal replacement. Eating behavior, eating traits and dietary preferences were evaluated by self-reporting via the Dutch Eating Behavior Questionnaire–Emotional Eating subscale (DEBQ-EE), the Dietary Fat and Free Sugar Questionnaire (DFS), and stress sensation via the Trier Inventory for Chronic Stress Screening Scale (TICS). Additionally, patients were asked to categorize their overall food intake after RYGB subjectively. Hunger, satiety and meal pleasantness were assessed via digital VAS. Extreme responders after RYGB donated stool sample for microbiota and metabolome. Pre-surgery clinical data were collected from the IFB Adiposity database and medical reports. See [34] for further descriptions of the study cohort.

### 4.2. Bariatric Surgery

Patients (*n* = 23) were operated at the certified section for bariatric surgery, department of Visceral, Transplant, Thoracic und Vascular Surgery at the University Hospital Leipzig between 2010 and 2015. The same bariatric surgeon performed all operations laparoscopically. Biliopancreatic limb length was 50 cm and Roux (alimentary) limb length was 150 cm, except in one patient from the bad responder group (80 cm and 170 cm, respectively). Moreover, in one patient (good responder) biliopancreatic limb length was unknown, and in another patient (good responder) esophagojejunostomy was performed instead of a gastric pouch due to incidentally discovered Barrett’s carcinoma. All patients received multi-disciplinary team assessment before surgery and were offered a structured four-year follow-up program with routine dietician, physician, surgeon and psychologist visits.

### 4.3. Microbiome Analyses

Bacterial DNA content was isolated with QIAamp Stool Mini Kit (Qiagen) and V3-V4 variable regions of the 16S rRNA amplified by PCR. Next, paired-end 2 × 250 bp Illumina sequencing was used. Analyses were done by GENEWIZ Germany GmbH, Leipzig. Raw sequencing data were processed in fastq format using the DADA2 R software package [84]. Low-quality reads and noise were removed, paired ends joined, forward reads trimmed at base pair position 280, reverse reads trimmed at base pair position 200 and amplicon sequence variants (ASVs) constructed. The Ribosomal Database Project (RDP) database [85] and DADA2 were then used to assign taxonomy to ASVs. Normalization of ASV read counts and calculation of relative abundance for each taxonomic level was executed with R script Rhea. Further bioinformatics and visualization were done with in-house written R-scripts. Alpha-diversity indices and Beta-diversity were calculated using the vegan R-package [86]. Non-metric multidimensional scaling (NMDS) was used to analyze beta-diversity of samples and group differences calculated by PERMANOVA. Significant differences in alpha-diversity and relative abundance of taxa between groups were calculated using the Kruskal–Wallis test, followed by post-hoc pairwise statistical analysis using the Dunn’s test. *p*-values were corrected for multiple testing using the Benjamini–Hochberg method where appropriate (number of independent tests > 20) [87]. Figures were constructed using the ggplot2 R-package [88].

### 4.4. Mass Spectrometric Measurements

The AbsoluteIDQ Bile Acid Kit (Biocrates Life Sciences AG) was used for bile acid analyses. Liquid chromatography-mass spectrometry (LC-MS/MS) measurements were carried out by MRM acquisition on a Waters Acquity UPLC System and a QTRAP 5500 (AB Sciex). Data were processed with Analyst Software (1.6.2) and MetIDQ Software (Biocrates Life Sciences AG). For measurements of amino acids and amines, the AbsoluteIDQ p180 Kit (Biocrates Life Sciences AG) was used on a QTRAP mass spectrometer (MS) applying electrospray ionization (ESI) (ABI Sciex API5500Q-TRAP). After separation through a precolumn (Security Guard, Phenomenex, C18, 4 × 3 mm; Phenomenex) and hyphenated reverse phase column (Agilent, Zorbax Eclipse XDB C18, 3.0 × 100 mm, 3.5 µm), analytes were quantified by multi reaction monitoring (MRM) which was standardized by applying spiked-in isotopically labelled standards in positive and negative mode. For data processing, MetIQ software (Biocrates Life Sciences AG) was used. The isotope-labeled chemical derivatization method described by Han et al. [89] was modified for quantification of SCFA. SCFA were chromatographically separated on an Acquity UPLC BEH C18 column (1.7 μm) (Waters) using H_2_O (0.01% FA) and acetonitrile (0.01% FA). Analytes were quantified and identified by the scheduled MRM method.

### 4.5. Statistical Analyses

Statistical analyses were performed using GraphPad Prism 9, IBM SPSS Statistics 24, and Microsoft Excel (Microsoft Office Profession Plus 2016). Data are expressed as mean ± standard deviation (SD), or median and interquartile range (IQR) for parameters that are not normally distributed. VAS data are presented as median with IQR. Non-normally distributed group comparisons were performed with the Mann–Whitney U test. Normally distributed group comparisons were performed with the unpaired Student’s *t*-test. Significance of difference of principal component analyses (PCA) of metabolite profiles was calculated by PERMANOVA and group comparisons of single metabolites calculated by the Kruskal–Wallis test with pairwise post-hoc Dunn’s test. *p*-values were corrected for multiple testing using the Benjamini–Hochberg method where appropriate (number of independent tests > 20) [87].

## Figures and Tables

**Figure 1 metabolites-12-00417-f001:**
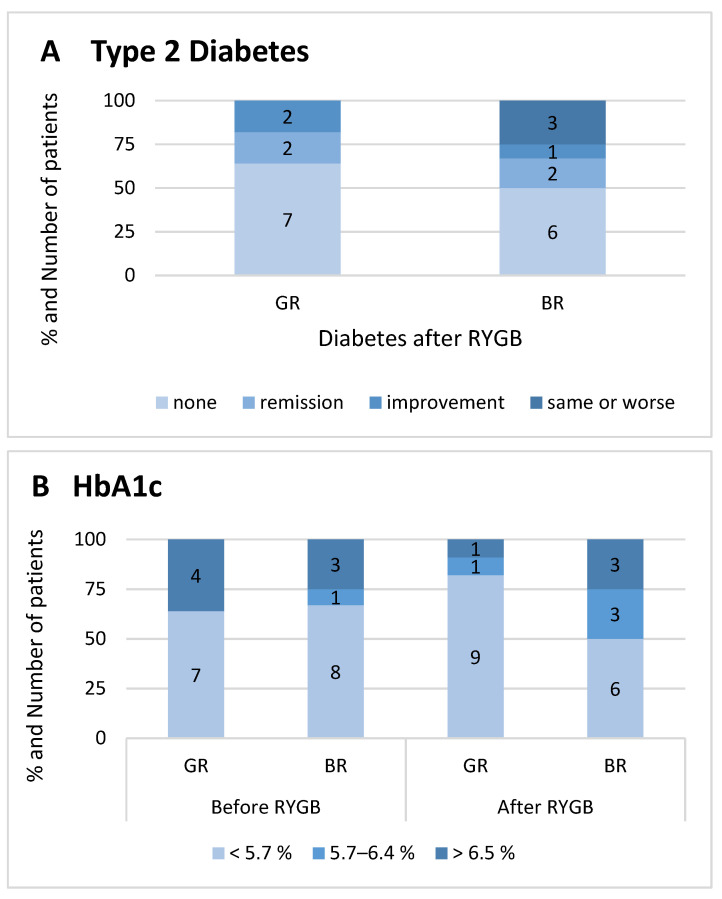
Relative and absolute proportion of patients in our cohort with and without (pre)diabetes. (**A**) Type 2 diabetes prevalence and remission status of both responder groups at follow up. (**B**) Hemoglobin A1c level independent of diabetes medication before RYGB intervention and at follow up. (**C**) Antidiabetic medication before RYGB intervention and at follow up. BR, bad responder group; GR, good responder group; HbA1c, hemoglobin A1c; RYGB, Roux-en-Y gastric bypass.

**Figure 2 metabolites-12-00417-f002:**
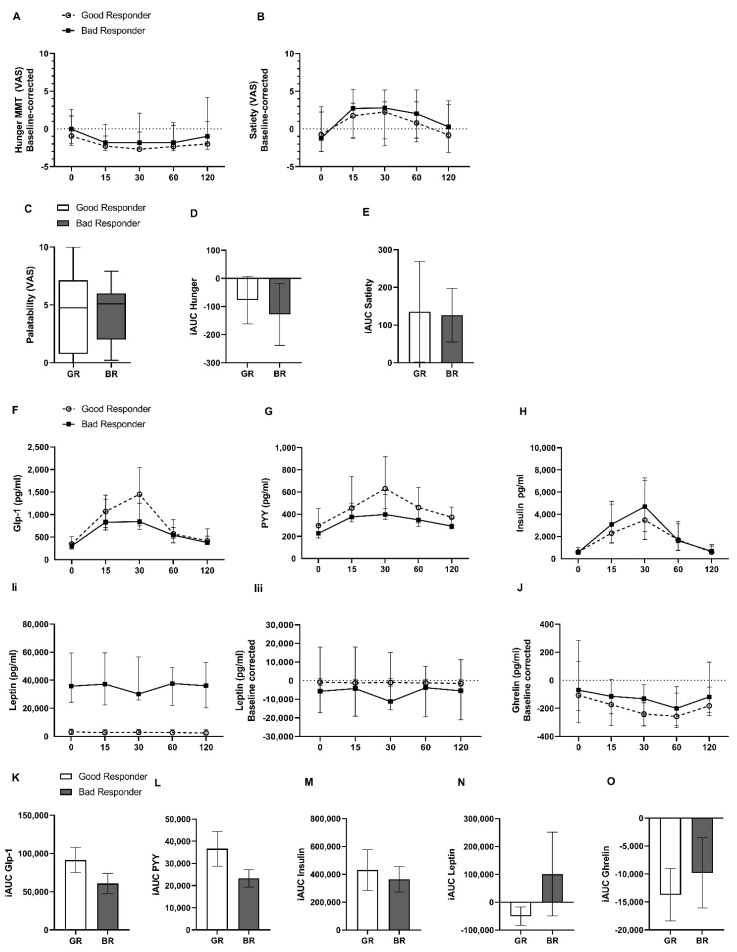
Visual analogue scale (VAS) data of hunger, satiety and palatability ratings (**A**–**C**) and gut hormone concentration (**F**–**J**) during standardized mixed-meal test at time points 15, 30, 60 and 120 min after ingestion of a test meal in the good and bad responder groups. Leptin and ghrelin levels (**G**,**H**) shown after baseline correction to account for weight-dependent effects. Data presented as median ± IQR (**A**,**B**,**D**–**H**); Boxplot of Palatability (whiskers 1.5 × IQR). Incremental areas under the curve (iAUC) ± S.E.M. for hunger and satiety and gut hormones (**K**–**O**).

**Figure 3 metabolites-12-00417-f003:**
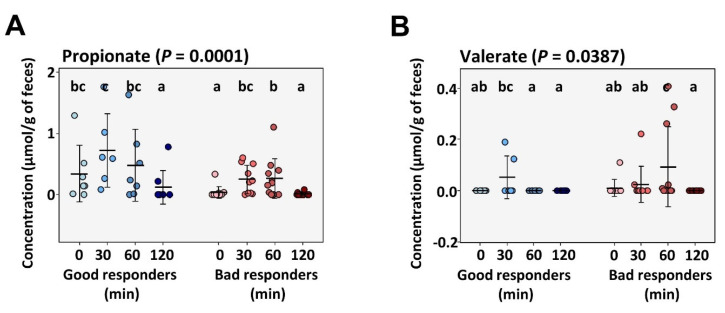
Circulating metabolites at fasted state and in response to a standardized meal test in patients with extreme weight loss response to RYGB. Three short chain fatty acids were quantifiable (propionate, valerate and acetate). Concentration of propionate (**A**) and valerate (**B**) are illustrated before (0 min) and 30, 60 and 120 min after ingestion of the liquid test meal with significance values calculated by Kruskal–Wallis test with pairwise post-hoc Dunn’s test. Values with different letters (a, b, c) are significant to each other. Beta-diversity based on principal component analysis of amino acid and biogenic amine pool in good and bad responders (**C**,**D**) after RYGB. Significance calculated by PERMANOVA. (**E**) depicts relative abundance (z-scores) of amino acids and biogenic amines measured before (0 min) and 30, 60 and 120 min after ingestion of the liquid test meal.

**Figure 4 metabolites-12-00417-f004:**
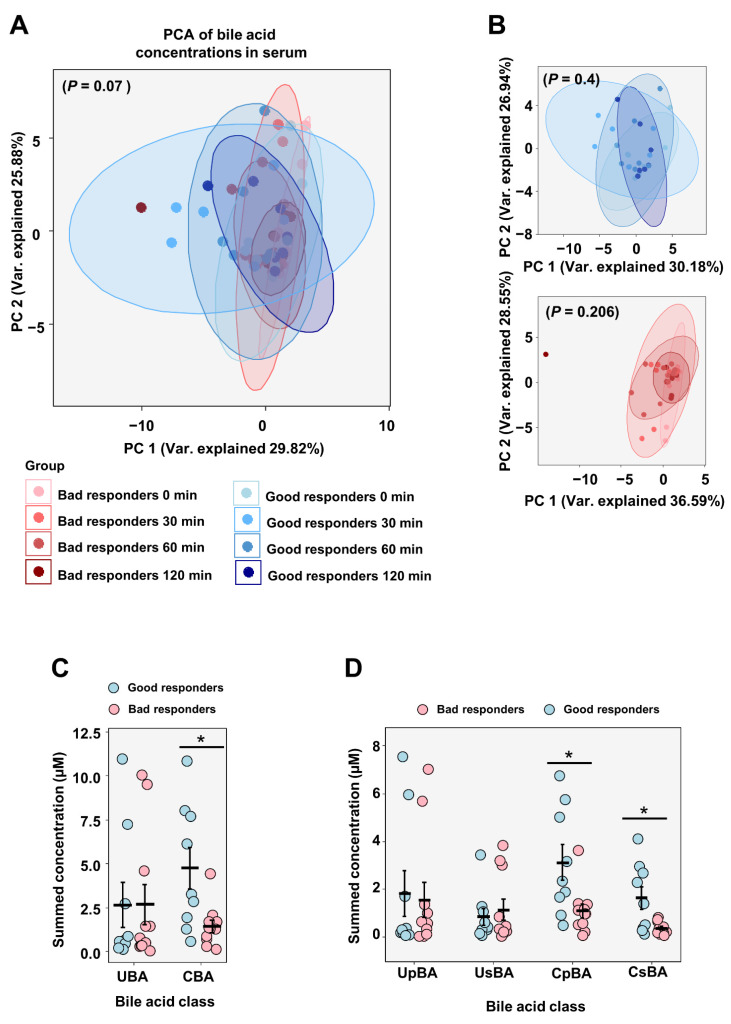
Serum bile acid concentrations in extreme weight loss responders to RYGB surgery. Principal component analyses of serum bile acid levels in good and bad responders (**A**,**B**) after RYGB surgery. Significance calculated by PERMANOVA (**C**) shows total unconjugated and conjugated fasting bile acid concentrations in good and bad responder groups which were further divided into primary and secondary bile acid species (**D**). Individual fasting bile acid concentrations shown in (**E**,**F**). Data shown as mean ± SEM. Group comparisons were performed with Student’s *t*-test. The symbol * indicates a *p*-value ≤ 0.05. CBA, conjugated bile acids; CpBA, conjugated primary bile acids; CsBA, conjugated secondary bile acids; UBA, unconjugated bile acids; UpBA, unconjugated primary bile acids; UsBA, unconjugated secondary bile acids; CA, cholic acid; CDCA, chenodeoxycholic acid; α-MCA, α-Muricholic acid; β-MCA, beta-Muricholic acid; DCA, Deoxycholic acid; HDCA, Hyodeoxycholic acid; LCA, Lithocholic acid; ω-MCA, ω-Muricholic acid; UDCA, Ursodeoxycholic acid; GCA, glycocholic acid; GCDCA, glycochenodeoxycholic acid; TCA, Taurocholic acid; TCDCA, Taurochenodeoxycholic Acid; TMCA (α+β), Tauro-muricholic acid; GLCA, glycolithocholic acid; GUDCA, Glycoursodeoxycholic acid; TDCA, Taurodeoxycholic acid; TLCA, Taurolithocholic acid; TUDCA, Tauroursodeoxycholic acid.

**Figure 5 metabolites-12-00417-f005:**
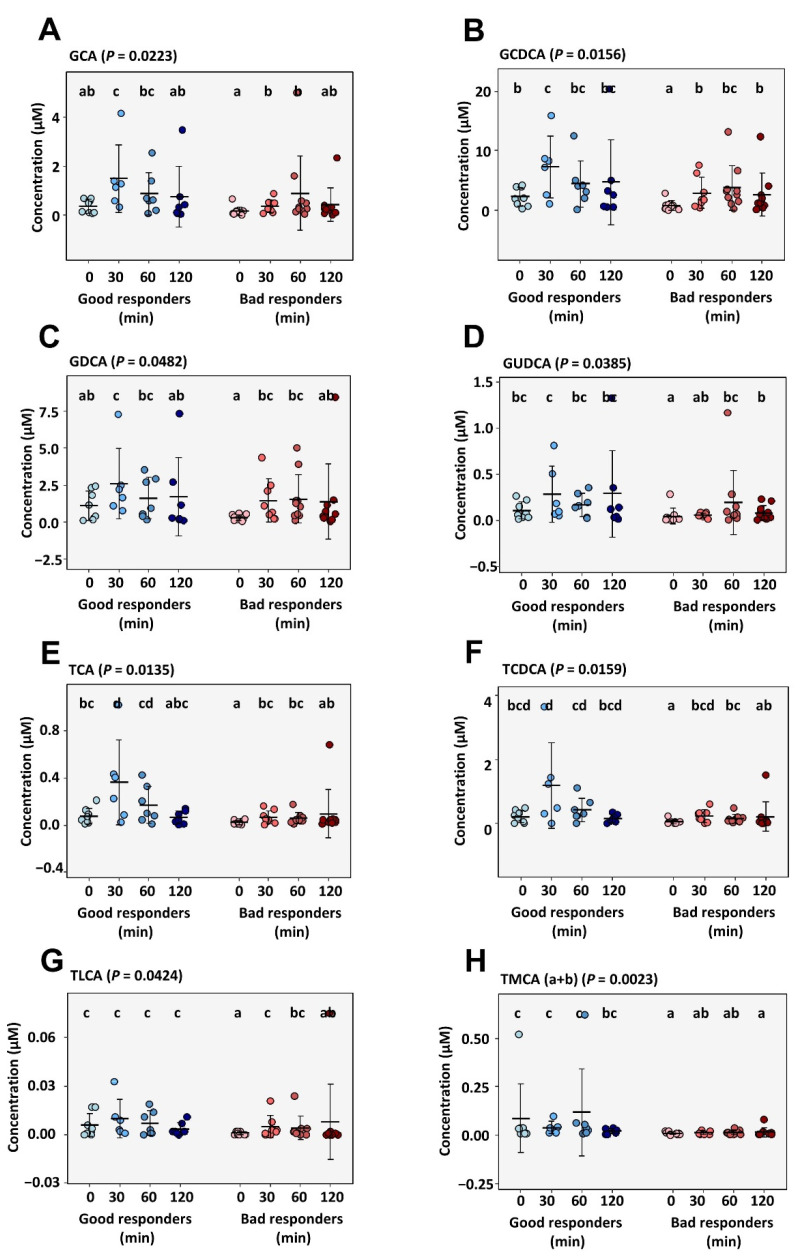
Serum bile acid levels in response to a standardized meal test after 0, 30, 60 and 120 min of ingestion in good and bad responders. (**A**–**H**) depicts concentrations of individual bile acid species measured in serum. Significance calculated by Kruskal–Wallis test with pairwise post-hoc Dunn’s test. Significance calculated by Kruskal–Wallis set with post-hoc pairwise Dunn’s test. Different letters (a–d) between values signify significant differences. GCA, glycocholic acid; GCDCA, glycochenodeoxycholic acid; GDCA, Glycodeoxycholic acid; GUDCA, Glycoursodeoxycholic acid; TCA, Taurocholic acid; TCDCA, Taurochenodeoxycholic Acid; TLCA, Taurolithocholic acid; TMCA (α+β), Tauro-muricholic acid.

**Table 1 metabolites-12-00417-t001:** Anthropometric data, eating behavior and metabolic profile of the study cohort (*n* = 23).

	Good Responder *n* = 11		Bad Responder *n* = 12		Good Responder vs. Bad Responder
**Clinical Characteristics**					
Sex (female/male)—*n* (%)	7 (64)/4 (36)		9 (75)/3 (25)		
Smokers—*n* (%)	4 (36)		2 (17)		
Diabetic—*n* (%) Before surgery (reported/A1c > 6.5%) After surgery (reported/A1c > 6.5%)	4 (36)/4 (36) 2 (18)/1 (9)		6 (50)/3 (25) 4 (33)/(3 (25)		
	**mean/median ***	**SD/IQR ***	**mean/median**	**SD/IQR**	***p*-value**
Education Years—yr	13	13–15.5	13	13––13	0.26
Age at test date—yr	52.9	± 9.5	54.1	± 10.6	0.78
Time after surgery—yr	4.3	± 1.2	4.6	± 1.5	0.60
Excess Weight Loss (EWL) at test date—%	93.0	± 4.3	19.5	± 13.3	**<0** **.0001**
Body weight—kg Before surgery Nadir At test date Change from baseline	133.4 69.9 76.6 −56.8	± 22.4 ± 5.8 ± 7.6 ± 18.6	145.7 110.4 130.3 −15.5	± 19.5 ± 15.9 ± 17.6 ± 10.9	0.17 **<0.0001** **<0.0001** **<0.0001**
BMI—kg/m^2^ Before surgery Nadir At test date Change from baseline	46.5 24.4 26.6 −19.8	± 7.5 ± 1.8 ± 1.2 ± 6.7	52.7 39.9 47.2 −5.5	± 6.6 ± 4.4 ± 6.4 ± 3.9	**0.04** **<0.0001** **<0.0001** **<0.0001**
**Questionnaire Scores at test date**					
Fat and Sugar Intake (DFS-Q All) Fat Sugar Fat and Sugar	50 23 9.8 16.3	± 10 ± 3.9 ± 3.4 ± 4.7	47 24 10.7 12.8	± 6.1 ± 4.6 ± 4.0 ± 4.3	0.43 0.71 0.59 0.09
Emotional Eating (DEB-Q-EE)	1.5	1.0–2.2	2.0	1.1–2.9	0.54
Chronic Stress (TICS)	14	10–19	15	7.3–18	0.91
**Metabolic Profile**					
Alanine transaminase—μkat/L Before surgery At test date Change from baseline	0.42 0.40 −0.05	0.33–0.92 0.29–0.47 −0.49 to 0.03	0.56 0.34 −0.2	0.38–1.1 0.31–0.66 −0.64 to −0.02	0.48 0.76 0.28
γ-glutamyl transferase—μkat/L Before surgery At test date Change from baseline	0.40 0.21 −0.17	0.28–0.70 0.13–0.42 −0.33 to −0.1	0.53 0.41 −0.12	0.360.79 0.25–1.20 −0.20 to 0.32	0.52 **0.05** 0.16
Fasting Glucose—mmol/L Before surgery At test date Change from baseline	5.7 5.0 −0.62	4.9–7.3 4.6–5.3 −2.3 to −0.1	6.5 5.4 −0.31	5.2–11 5.2–8.5 −2.5 to 0.03	0.24 **0.02** 0.69
Hemoglobin A1c—% Before surgery At test date Change from baseline	5.6 5.0 −0.32	5.0–7.0 4.8–5.6 −1.56 to −0.07	5.6 5.7 −0.4	5.3–8.5 5.2–6.6 −1.79 to 0.32	0.55 **0.03** 0.99
Insulin Resistance (HOMA-IR) Before surgery At test date Change from baseline	4.9 1.6 −2.6	2.4–18 1.1–2.5 −13.5 to −0.7	6.3 2.7 −3.3	3.9–11 1.8–3.1 −6.2 to −1.2	0.56 0.09 0.99
Triglycerides—mmol/L Before surgery At test date Change from baseline	1.3 0.94 −0.4	0.97–1.7 0.63–1.2 −0.64 to −0.08	1.4 1.3 −0.17	0.95–3.3 1.2–1.3 −1.78 to 0.15	0.85 **0.02** 0.69
Low Density Lipoprotein—mmol/L Before surgery At test date Change from baseline	2.5 2.2 −0.29	± 0.57 ± 0.54 ± 0.62	2.9 2.5 −0.48	± 1.3 ± 0.57 ± 0.93	0.37 0.28 0.58
High Density Lipoprotein—mmol/L Before surgery At test date Change from baseline	0.93 1.5 0.6	± 0.23 ± 0.33 ± 0.27	1.1 1.5 0.28	0.26 ± 0.33 ±0.3	0.09 0.70 **0.02**
C-reactive Protein—mg/L Before surgery At test date Change from baseline	5.0 0.3 −3.5	1.6–7.9 0.3–0.3 −7.6 to −1.3	9.7 2.0 −5.0	4.9–20 1.3–13 −9.9 to 2.0	0.10 **<0.0001** 0.79

* Data are presented as median and interquartile range (IQR) for non-normally distributed parameters and mean ± standard deviation (SD) for normally distributed parameters. Values with range are median and IQR, plus–minus values are means ± SD. Significant values *p* < 0.05 are printed in bold. Laboratory values are expressed in SI-values. *Questionnaires:* Dutch Eating Behaviour Questionnaire–Emotional Eating subscale (DEBQ-EE); Dietary Fat and Free Sugar Questionnaire ((DFS); Trier Inventory for Chronic Stress Screening Scale (TICS). BMI, body mass index; CRP, C-reactive protein; excess weight loss; HOMA-IR, homeostasis model assessment for insulin resistance; kg, kilograms; yr, years.

## Data Availability

The authors confirm that the data supporting the findings of this study are available within the article and its Appendix A.

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
