# Peer review of "Metabolic Profile and Metabolite Analyses in Extreme Weight Responders to Gastric Bypass Surgery"

_metabolites, 2022, doi:10.3390/metabo12050417_

Round 1

Reviewer 1 Report

The purpose of the study was quite ambitious: ‘The current study was designed to profile the composition of the gut microbiota and  the circulating metabolomic signature in patients with variable weight loss response to RYGB surgery and to determine associations with weight loss-independent beneficial metabolic outcomes after surgery’.

The authors have put a lot of effort into achieving the intended goal but the only conclusion they managed to draw was: ‘Patients with divergent weight loss response to RYGB surgery show distinct modulations of both fasted and postprandial bile acid profiles.’

As hypothesized by the same authors, the increase in serum bile acids could activate some nuclear membrane receptors, in particular FXR, which are able to modulate lipid metabolism at different levels. Despite this is a rather interesting result, the paper has some small or large biases.

From what this referee seems to understand, the authors measured the concentration and pattern of serum bile acids before and after a standardized meal. It does not seem that they have measured the pool as they have stated somewhere in their paper (195, 501 lines, figure 4. They need to clarify this point.

Higher fasting and postprandial serum bile acids were found in good responders. The authors comment: ‘After ingestion of the test meal, good responders showed a clearly higher secretion’. The use of the term secretion seems improper in this respect. Peripheral concentrations of serum BA are the result of balance between the intestinal input and the hepatic clearance from the portal blood. In the presence of a normal liver the intestinal input is the major determinant of serum BA concentrations In liver disease decreased hepatic clearance is assumed to play a role of major importance. Indeed, diminished hepatic extraction and presence of extrahepatic shunting may result in increased systemic delivery of BA and elevated serum BA levels, despite a decrease in hepatic synthesis.

What the authors did not notice is that in the serum of their patients bile acids were predominantly conjugated while in healthy subjects unconjugated bile acids are the majority. It is not clear what this difference is due to. Can they speculate on that?

Why do the Authors use for the whole group of examined patients the term Extreme weight loss responders?

More generally, the authors recognize some limitations of their work:’ small sample size, no longitudinal design and assessment of eating behavior, diet and exercise by self-reporting measures’.

The results are described in great detail, perhaps too much, and the discussion is far too long. All the paper should be very shortened.

Author Response

Please find our point-by-point response in the attached word document.

Reviewer 2 Report

This paper deals with an interesting and relevant topic.

I suggest that the paper may be published in Metabolites, provided the following topics are being covered by the authors:

  • Please include and use abbreviation for type 2 diabetes, T2D (starting at line 4 introduction), in alignment with other abbreviations used.
  • The figures are not self explanatory and should be improved as such. For example, Fig 3: it is from the figure and corresponding legend itself unclear what is actually displayed. 0-30-60-90 min please explain briefly, same for a , bc, etc.
  • Regarding the PCA figure 3A. Please include a graph displaying the PCA performed on both good and bad responders altogether. It is important to see whether good/bad responders can be separated, which is not possible to see here because the principal components for PCA done on respectively good or bad responder will differ (loading of PC's will be different for both groups of patients).
  • The authors claim in the discussion that it is one of the most challenging clinical issues to predict responders from non responders. As such, it is surprising that no baseline microbiota profile of the patients is being measured (so microbiota profile before RYGB), because it is known from earlier studies using FMT to improve insulin sensitivity for metabolic syndrome patients to play a decisive role: only patients with a high baseline microbial diversity did respond to the FTM. This should be discussed, and please include the reference as well: Kootte, Ruud S., et al. "Improvement of insulin sensitivity after lean donor feces in metabolic syndrome is driven by baseline intestinal microbiota composition." Cell metabolism 26.4 (2017): 611-619.
  • Also, considering the fact that the authors discuss possible factors that explain the responder / non responder ratio, please include a discussion on intrinsic interindividual and intraindividual variation which may be pronounced and influences this ratio. Please discuss and include two recent references on this:
    • Larsen, O. F. A., E. Claassen, and Robert Jan Brummer. "On the importance of intraindividual variation in nutritional research." Beneficial microbes 11.6 (2020): 511-517.
    • Olsson, Lisa M., et al. "Dynamics of the normal gut microbiota: A longitudinal one-year population study in Sweden." Cell Host & Microbe (2022).

Author Response

Please find our point-by-point response in the attached word document. Thank you.

Reviewer 3 Report

Reviewers comment

In the study titled ‘Metabolic Profile and Metabolite Analyses in Extreme Weight Responders to Gastric Bypass Surgery, the authors Fries et al. attempted to find the reason for some individuals responded poor after Roux-en-Y gastric bypass (RYGB). The authors analyzed gut biome composition, bile acid, and its metabolites from the stool and blood of the patients who were extreme responders to RYGB. Microbiome analysis was done by sequencing. Bile acid and its metabolites were analyzed by LC-MS/MS. A total of 23 patients were recruited as extreme responders for the study and grouped as good and bad responders. 

The study found no difference in hunger and satiety sensation, in fasting or postprandial gut hormone release, gut microbiota composition, and metabolic outcome between both groups. Out of many metabolites analyzed in stool and blood, a higher quantity of propionate in good and valerate in bad responders was observed. Conjugated primary and secondary bile acids were higher in good responders in the fasted and postprandial state. Poor responders are with higher BMI.

Obesity beyond adipose tissue is poorly understood. This study is an attempt in that direction. Although the manuscript is written well and the concept is good, the conclusions derived from the small sample size for this kind of prolonged study may deviate from the actual results. Hence the authors may consider including the limitations of the study.

Author Response

Reviewer 3:

In the study titled ‘Metabolic Profile and Metabolite Analyses in Extreme Weight Responders to Gastric Bypass Surgery, the authors Fries et al. attempted to find the reason for some individuals responded poor after Roux-en-Y gastric bypass (RYGB). The authors analyzed gut biome composition, bile acid, and its metabolites from the stool and blood of the patients who were extreme responders to RYGB. Microbiome analysis was done by sequencing. Bile acid and its metabolites were analyzed by LC-MS/MS. A total of 23 patients were recruited as extreme responders for the study and grouped as good and bad responders. 

The study found no difference in hunger and satiety sensation, in fasting or postprandial gut hormone release, gut microbiota composition, and metabolic outcome between both groups. Out of many metabolites analyzed in stool and blood, a higher quantity of propionate in good and valerate in bad responders was observed. Conjugated primary and secondary bile acids were higher in good responders in the fasted and postprandial state. Poor responders are with higher BMI.

R1. Obesity beyond adipose tissue is poorly understood. This study is an attempt in that direction. Although the manuscript is written well and the concept is good, the conclusions derived from the small sample size for this kind of prolonged study may deviate from the actual results. Hence, the authors may consider including the limitations of the study.

A1. We thank the reviewer for this concise summary of our study. We agree with the reviewer, that the main limitations of our study are firstly the small sample size, that essentially limit application of our results to a broad spectrum of patients, and secondly, the cross-sectional design. We have elaborated on this in more detail in the revised manuscript (see line 340-348). Thank you.

Round 2

Reviewer 2 Report

To my opinion, this paper can now be published in its present form.

Reviewer 3 Report

The changes made to the manuscript are satisfactory.